# ACDC: Autoregressive Coherent Multimodal Generation using Diffusion Correction

## Abstract

Autoregressive models (ARMs) and diffusion models (DMs) represent two leading paradigms in generative modeling, each excelling in distinct areas: ARMs in global context modeling and long-sequence generation, and DMs in generating high-quality local contexts, especially for continuous data such as images and short videos. However, ARMs often suffer from exponential error accumulation over long sequences, leading to physically implausible results, while DMs are limited by their local context generation capabilities. In this work, we introduce Autoregressive Coherent multimodal generation with Diffusion Correction (ACDC), a zero-shot approach that combines the strengths of both ARMs and DMs at the inference stage without the need for additional fine-tuning. ACDC leverages ARMs for global context generation and memory-conditioned DMs for local correction, ensuring high-quality outputs by correcting artifacts in generated multimodal tokens. In particular, we propose a memory module based on large language models (LLMs) that dynamically adjusts the conditioning texts for the DMs, preserving crucial global context information. Our experiments on multimodal tasks, including coherent multi-frame story generation and autoregressive video generation, demonstrate that ACDC effectively mitigates the accumulation of errors and significantly enhances the quality of generated outputs, achieving superior performance while remaining agnostic to specific ARM and DM architectures. Project page: https://acdc2025.github.io/

## 1 Introduction

Autoregressive models (ARM) (Brown, 2020) and diffusion models (DM) (Ho et al., 2020) are the two prominent paradigms in modern generative modeling, each with its strengths, weaknesses, and areas where they excel. ARMs excel at modeling discrete token sequences, leading to their domination of the language domain (Dubey et al., 2024; Abdin et al., 2024), and more recently, multimodal modeling (Kondratyuk et al., 2023; Lu et al., 2024; Xie et al., 2024; Jin et al., 2024), as vector quantized (VQ) autoencoder architectures become advanced (Esser et al., 2021; Yu et al., 2023b). ARMs are useful for 1) modeling the *global* context, as it attends causally to all previous tokens, and 2) *long-sequence* generation, as they are not limited to fixed-length tokens (Liu et al., 2023; 2024). These advantages render ARMs excellent for high-level long-term planners (Du et al., 2023; Huang et al., 2024a) and world models (Ha & Schmidhuber, 2018; Hao et al., 2023; Ge et al., 2024b). However, the "plans" generated by ARMs cannot be corrected, often leading to physically implausible results (Kambhampati et al., 2024) and low quality images/videos. To mitigate this issue, one often needs an external verifier to locally correct for the mistakes (Kambhampati et al., 2024) and rely on vision-specific decoder fine-tuning (Ge et al., 2024b;a). The inability to correct for the errors exacerbate as the length of the generated sequences becomes longer, inherently due to the exponential error accumulating nature of these models. The accumulated errors are especially noticeable in autoregressive video generation, with videos often diverging when generating over a few tens of frames (Liu et al., 2024).

On the other hand, DMs excel at modeling continuous vectors, especially capable of generating high-quality images (Rombach et al., 2022; Saharia et al., 2022) and short video clips (Brooks et al., 2024). The generative process in diffusion models, which transitions from noise to the data space, inherently creates a Gaussian scale-space representation of the data distribution. This approach allows the model

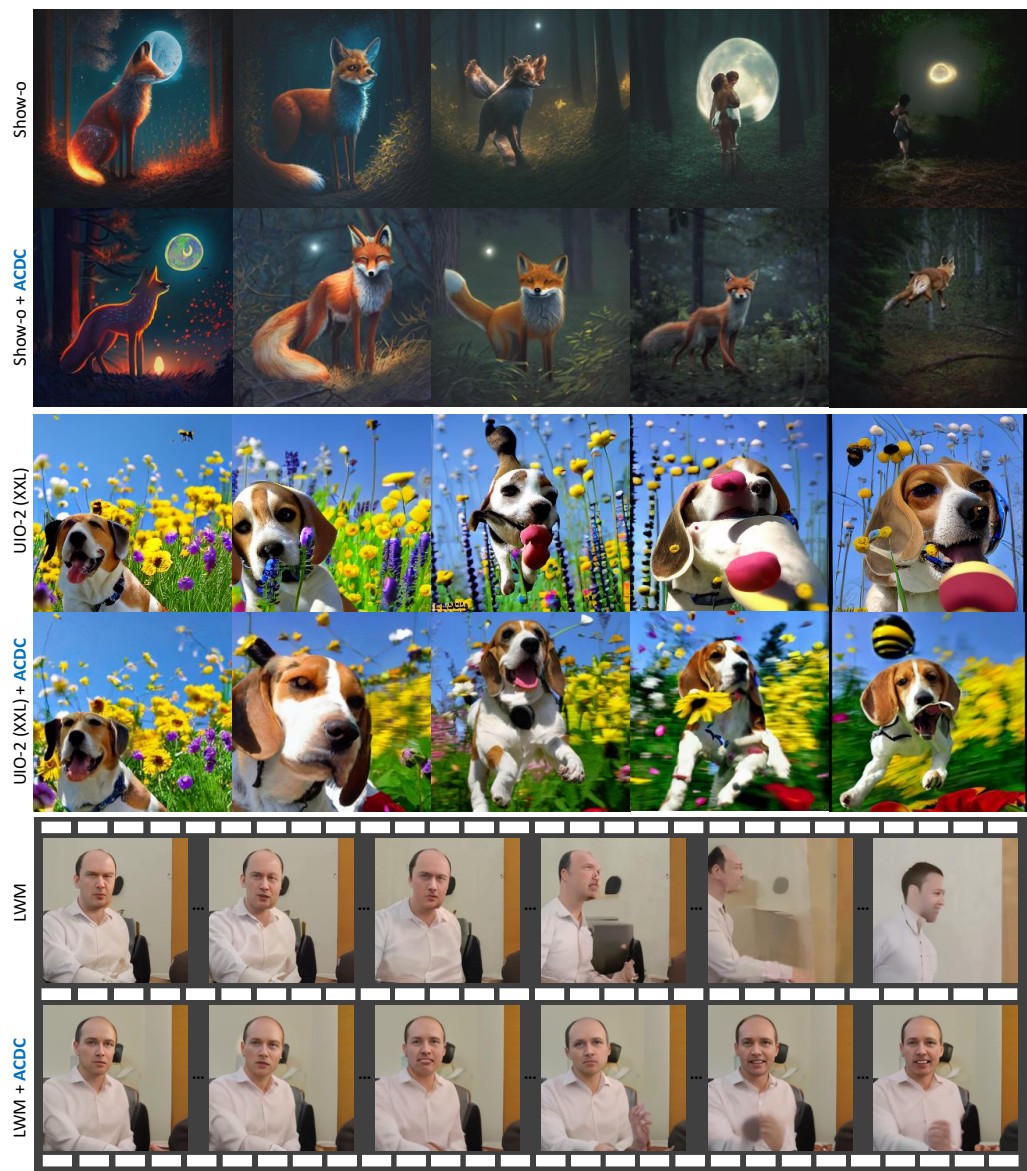

Figure 1: Multimodal ARM and its ACDC corrected version. Full prompts: App. F
Row 1-2: (**Story generation**) "Fox in moonlit forest... slip through underbrush ... leaped over a log"
Row 3-4: (**Story generation**) "Beagle in a garden with blooming flowers ... jumped ... barked"
Row 5-6: (**Autoregressive video generation**) "A male interviewer listening to a person talking".

to iteratively refine the generated image in a coarse-to-fine manner, capturing both global structure and fine-grained details effectively. In other words, DMs are excellent *local* context generators. Another intriguing feature of diffusion models (DMs) is their high degree of controllability. A pre-trained DM can be repurposed for a variety of tasks, such as solving inverse problems (Chung et al., 2023), image editing (Meng et al., 2021; Kim et al., 2022), adversarial purification (Nie et al., 2022), and more, by reconstructing the inference path and applying appropriate guidance. Since DMs function as score estimators, they implicitly serve as priors for the data distribution (Song & Ermon, 2019), ensuring that generated or modified samples remain close to the high-density regions of the distribution. However, DMs typically generate fixed-length vectors, which limits their suitability for tasks involving long sequence generation or planning.

Combining the advantages of ARMs and DMs, researchers have investigated ways to combine the two to by using ARM as the global model and DM as the local vision generator. This led to the construction of multimodal generators (Ge et al., 2024a; Li et al., 2024), robot planners (Du et al., 2023), and world models (Xiang et al., 2024). However, all these models require re-purposing of the base models, and require expensive modality-specific fine-tuning or instruction tuning. Moreover, to the best of our knowledge, using DMs as a way to avoid exponential error accumulation of ARMs has not been investigated in the literature.

To fill in this gap, in this work, leveraging the intriguing strengths of both pre-trained ARMs and DMs, we present Autoregressive coherent multimodal generation with Diffusion Correction (ACDC), a flexible **zero-shot** combination method of existing models applicable at inference stage, regardless of the specific design of the ARMs and DMs. In ACDC, the ARM is the main workhorse, responsible for generating multimodal tokens that respect the global context. For every chunk of vision tokens generated (e.g. a single image, 16 video frames), the tokens are decoded into the continuous space with the visual detokenizer, and locally corrected with a *memory-conditioned* DM through the SDEdit (Meng et al., 2021) process. Specifically, to distill the global context of the generated sequence into the DM without any fine-tuning, we additionally propose a memory module implemented with a large language model (LLM), which causally corrects the conditioning texts of the DM.

Through extensive experiments with various multimodal ARMs including Large World Models (Liu et al., 2024), Unified-IO-2 (Lu et al., 2024), and Show-o (Xie et al., 2024) on two distinct tasks: coherent multi-frame story generation and autoregressive video generation, we show that ACDC significantly improves the quality of the generated results and can correct for the exponential accumulation in errors (See Fig. 1 for representative results), achieving all this while being training-free and agnostic to the model class.

## 2 RELATED WORKS

### 2.1 UNIFYING AUTOREGRESSIVE MODELS WITH DIFFUSION MODELS

**Diffusion models as vision decoder for autoregressive models** Instead of the standard way of using VQGAN decoder as the visual detokenizer, works such as SEED-X (Ge et al., 2024a) and Mini-Gemini (Li et al., 2024) propose to leverage Stable Diffusion XL (SDXL) (Podell et al., 2023) as the visual decoder by fine-tuning a pre-trained SDXL to take in the visual tokens as the condition through attention. While the image generation quality of these methods exceed the ones that only use ARMs since the pre-trained diffusion model guarantees high-quality outputs, consistency is hard to control. Pandora (Xiang et al., 2024) is a work that shares a goal that is similar to ours, which uses two Q-formers (Li et al., 2023a) as adapters to connect ARMs and DMs, repurposing them as a world model.

**Proposals in unified architecture** For methods that use DMs as vision decoders, the ARM component is still responsible for generating the multimodal tokens, and hence the methods can be considered as late-fusion methods. Recently, several methods have been proposed to combine the two model classes in the earlier pre-training stage in an early-fusion manner. Transfusion (Zhou et al., 2024) trains the LMM with the next token prediction loss for language and with the diffusion (i.e. denoising) loss for image patches. The sampling modes are switched between next token prediction and continuous diffusion when the special token is sampled. Show-o (Xie et al., 2024) is similar to Transfusion in that it uses both next token prediction and diffusion, but differs in that the discrete tokens are generated through discrete diffusion (Austin et al., 2021; Yu et al., 2023a). Diffusion forcing (Chen et al., 2024a) generalizes ARMs so that it can take perform *noisy* autoregression, as well as iteratively refine the current token by denoising it.

While promising, all of the methods discussed require altering the architecture of the DM[1] or the ARM, as well as extensive fine-tuning and aligning. Our goal, on the other hand, is to leverage the pre-trained model components *as is*, without any fine-tuning. For example, later we show that Show-o also benefits from the use of the proposed ACDC.

---

[1]For instance, conditioning the diffusion model on the previous frame (Ge et al., 2024a) or video (Xiang et al., 2024)

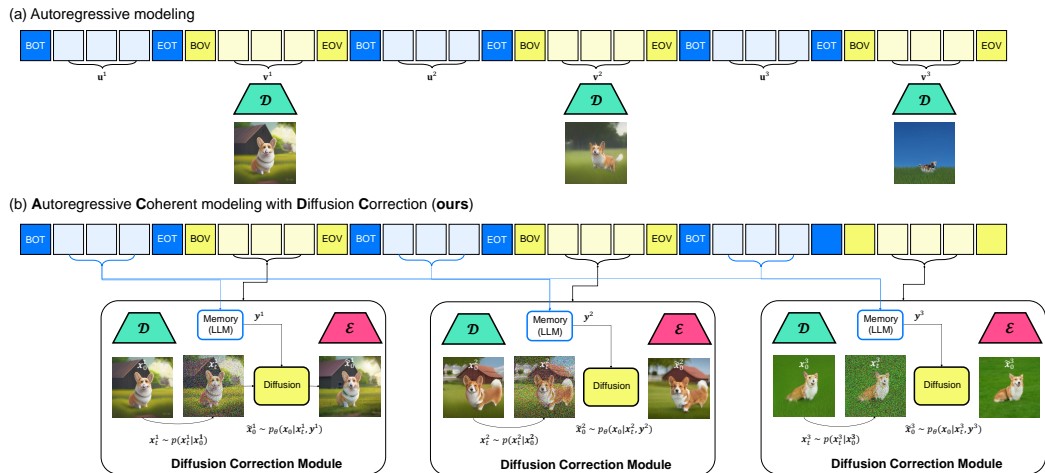

Figure 2: Illustration of the proposed ACDC method. For each chunk of image token $v^i$ decoded into a a frame $x_0^i$, 1) LLM memory module causally summarizes the previous prompts $y^{1:i-1}$ to provide the text input $y^i$, 2) Diffusion correction is applied by perturbing $x_0^i$ with forward diffusion, then running reverse diffusion conditioned on $y^i$. {B,E}o{T,V} indicates beginning, end of text, vision tokens, respectively.

## 2.2 DIFFUSION MODELS FOR IMAGE/VIDEO SEQUENCE GENERATION

**Diffusion models as world models**   Recently, several works aimed to repurpose diffusion models as world models (Ha & Schmidhuber, 2018), which estimate the next state (i.e. image or video) given the current state and action. GenHowto (Damen et al., 2024) fine-tuned a DM to predict the next image frame on the previous frame and a text instruction on a targetted dataset (Souček et al., 2022). SEED-story (Yang et al., 2024) achieves a similar goal on another targetted story dataset but on a longer sequence. Genie (Bruce et al., 2024) and GameNGen (Valevski et al., 2024) take latent actions (e.g. control signal) as input to output the game frame. Going further, world models that can generate chunks of video frames (Xing et al., 2024; Xiang et al., 2024) were also proposed. While promising as world models, these models do not possess the understanding and planning abilities of LMM, and again, deviate from a general-purpose DM.

**Diffusion models as long sequence generators**   Standard video diffusion models (Ho et al., 2022; Blattmann et al., 2023; Brooks et al., 2024) generate data of fixed length. One can repurpose video DMs for arbitrary-length video generation by using diagonal denoising proposed in FIFO-diffusion (Kim et al., 2024), or train a video DM from scratch using a similar strategy (Ruhe et al., 2024). Unfortunately, even the models that are repurposed for long sequence generation can only attend to a local context, hampering the consistency of the generated videos.

## 3 AUTOREGRESSIVE COHERENT MODELING WITH DIFFUSION CORRECTION

In this section, we explore the use of Multimodal ARMs as world models used in the task of coherent story and video generation, where the key lies in the fidelity of the model's predictions, as well as controlling the exponential accumulation of errors. Specifically, we assume a Partially Observable Markov Decision Process (POMDP) setup (Sutton, 2018) where the actions are described by texts and the states are rendered as image frames.

**Autoregressive multimodal modeling for world models**   For predictive planning (Chang et al., 2020; Zhao et al., 2022; Damen et al., 2024), given the initial state and the goal, both a policy and a world model is required. The policy is responsible for predicting the next action from the current state. The world model, on the other hand, should be capable of modeling the next state from the current state and action pair.

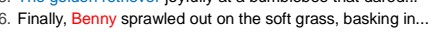
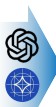

Figure 3: Before (left) and after (right) correction through the proposed LLM memory module. Key global context is distilled into the local prompts.

Let us define $\mathbf{u}^i$ as a chunk of $i$-th text tokens with sequence length $m$, and $\mathbf{v}^i$ as a chunk of image tokens with sequence length $n$. In a multimodal multi-turn generation setup, at the $i$-th turn of generation of the image keyframe, as illustrated in Fig. 2, the base ARM will sample from

$$\mathbf{v}^i \sim p_\phi(\mathbf{v}^i|\mathbf{v}^{1:i-1}, \mathbf{u}^{1:i}), \quad \text{where} \quad \mathbf{v}^{1:i-1} := [\mathbf{v}^1, \mathbf{v}^2, \cdots, \mathbf{v}^{i-1}], \mathbf{u}^{1:i} := [\mathbf{u}^1, \mathbf{u}^2, \cdots, \mathbf{u}^i]. \quad (1)$$

where the ARM is parameterized with $\phi$. For the image tokens, the chunk can be decoded autoregressively in a patch-wise raster scan order, or through a discrete diffusion process (Yu et al., 2023a). Here, $\mathbf{u}^i, \mathbf{v}^i$ are *discrete* tokens. To visualize, the discrete image tokens are decoded to the continuous pixel space with a visual detokenizer (e.g. decoder of a VQ-GAN (Esser et al., 2021)) $\boldsymbol{x}_0^i = \mathcal{D}(\mathbf{v}^i)$. To make a distinction between the continuous decoded variable and the discrete token variable, note that we use bold italics for the decoded variables. Here, the subscript in $\boldsymbol{x}_t^i$ denotes the diffusion time[2], and the superscript denotes the physical time (index).

**Diffusion correction** The decoded $i$-th image $\boldsymbol{x}_0^i$ often contains visual artifacts. The problem is less significant in the initial frame but exacerbates for later predicted frames. To correct the visual artifacts, our goal is to **(1)** bring $\boldsymbol{x}_0^i$ closer to the clean data manifold, yet **(2)** proximal to the starting point. This can be effectively implemented with a text-conditional SDEdit (Meng et al., 2021), where one can use any text-to-image (T2I) DM, such as stable diffusion (Rombach et al., 2022) that operates in the latent space, or DeepFloyd IF (Team, 2023) that operates in the pixel space. As illustrated in Fig. 2, the process reads

$$\tilde{\boldsymbol{x}}_0^i \sim p_\theta(\boldsymbol{x}_0|\boldsymbol{x}_{t'}^i, \boldsymbol{y}), \quad \boldsymbol{x}_t^i \sim p(\boldsymbol{x}_t^i|\boldsymbol{x}_0^i), \quad (2)$$

where $\theta$ is the DM parameters, $t'$ is a hyperparameter that chooses the degree of perturbation, and $\boldsymbol{y}$ is the text condition. Typically, we choose a moderate scale of $t' \in [0.4, 0.5]$ such that the correction does not alter the content of the image, but only make local corrections. After obtaining $\tilde{\boldsymbol{x}}_0^i$, if this is not the terminal state, we can re-encode it back to the image tokens, i.e. $\tilde{\mathbf{v}}^i = \mathcal{E}(\tilde{\boldsymbol{x}}_0^i)$, where $\mathcal{E}$ is the image encoder of the ARM, and the sampling proceeds with the swapped tokens. In Appendix B, we theoretically show that our diffusion correction algorithm meets both criteria **(1)**—Theorem 1 and **(2)**—Theorem 2,3.

**Large Language Models as memory module** Care must be taken when choosing the text condition as input for our diffusion correction module $p_\theta$ due to two factors. First, the DM is a *local* image correction module that should only be conditioned for the current frame, and not the spurious information from the past states. However, it should also take into account the key information from the previous states, such as the example given in Fig. 3. Looking at the second frame only, it is impossible to deduce that "His" in the local context is referring to "Benny, the golden retriever". While one could aim to design and train a separate memory module specified for this task, we take a simpler approach, where we take a large language model (LLM) to causally generate new prompts conditioned on the previous prompts keeping the key information. Concretely, with an LLM parametrized with $\varphi$ and the system/user prompt $\mathbf{d}$ for the memory module, we change Eq. (2) to

$$\tilde{\boldsymbol{x}}_0^i \sim p_\theta(\boldsymbol{x}_0|\boldsymbol{x}_{t'}^i, \boldsymbol{y}^i), \quad \boldsymbol{x}_{t'}^i \sim p(\boldsymbol{x}_{t'}^i|\boldsymbol{x}_0^i), \quad \boldsymbol{y}^i \sim p_\varphi(\boldsymbol{y}^i|\boldsymbol{y}^{1:i-1}, \mathbf{d}), \quad (3)$$

where $\boldsymbol{y}^i$ denotes the $i-$th input prompt to the DM that is refined by the LLM memory module by summarizing $\boldsymbol{y}^{1:i-1}$, i.e. the previous prompts used.

Our algorithm is illustrated in Fig. 2. The importance of memory module can be seen theoretically in Theorem 3. We provide the implementation details of the memory module in Appendix D.

---

[2]We review diffusion models in Appendix A.

| Method | Frame consistency ($\uparrow$) | CLIP-sim ($\uparrow$) | ImageReward ($\uparrow$) | FID ($\downarrow$) |
|---|---|---|---|---|
| Stable Diffusion v1.5 (Rombach et al., 2022) | 0.6822 | 28.91 | $-0.8455$ | **32.11** |
| Show-o (Xie et al., 2024) | 0.8211 | 28.76 | $-0.5752$ | 60.50 |
| Show-o (Xie et al., 2024) + ACDC (ours) | **0.9062**▲10.4% | **30.82**▲7.16% | **$-0.0003$**▲0.574 | 56.36▲7.34% |
| UIO-2$_{\text{XXL}}$ (Lu et al., 2024) | 0.8833 | 29.46 | $-0.8354$ | 67.12 |
| UIO-2$_{\text{XXL}}$ (Lu et al., 2024) + ACDC (ours) | 0.8962▲1.46% | **31.09**▲5.53% | $-0.2624$▲0.574 | 57.80▲16.1% |

Table 1: Quantitative evaluation of the Story generation task. **Best**, second best.

**Incorporating physical constraints**   When generating images with multimodal ARMs, the degradation in the quality of images is not the only issue. Specifically, we observe that the ARMs are less sensitive to the *physical feasibility* of the rendered image. For instance, we often observe cases where rabbits have three ears, or where there is more than one moon or sun in the background, as can be seen in Fig. 5. Fortunately, incorporating physical constraints to diffusion models has been widely explored in the context of inverse problems (Chung et al., 2023; Yuan et al., 2023) and conditional diffusion models (Zhang et al., 2023). Thanks to the versatility of diffusion inference, one can additionally incorporate user feedback (e.g. inpainting masks, depth constraints) to explicitly correct for the errors, or automatically project the image to the manifold of feasible data (Gillman et al., 2024).

**Extension to autoregressive video generation**   We consider an extension to autoregressive video generation from a single text prompt using LWM (Liu et al., 2024). Note that as we use a multimodal ARM as our base model, extension to multi-prompt input between the frames is trivial. In an autoregressive text-to-video generation, sampling is done similar to Eq. (1)

$$\boldsymbol{x}_0^i = \mathcal{D}(\mathbf{v}^i), \quad \mathbf{v}^i \sim p_\phi(\mathbf{v}^i | \mathbf{v}^{1:i-1}, \mathbf{u}), \tag{4}$$

where $\mathbf{u}$ is the prompt, and the video frames are decoded independently with a VQ-GAN decoder. When generating videos using AR models, visual artifacts often appear in the generated frames, and inconsistencies in motion between these frames may occur due to the independent decoding. To resolve these issues, we propose to use our diffusion correction scheme analogous to Eq. (2) with text-to-video (T2V) DMs, such as AnimateDiff (Guo et al., 2024) or VideoCrafter2 (Chen et al., 2024b). Out of $N$ total frames to be generated with the ARM, let $L < N$ be the number of frames that T2V models take as input. After sequentially sampling the video frames to get $\boldsymbol{X}_0^{1:L} = [\boldsymbol{x}_0^1, \cdots, \boldsymbol{x}_0^L]$ we apply SDEdit similar to Eq. (2)

$$\tilde{\boldsymbol{X}}_0^{1:L} \sim p_\theta(\boldsymbol{X}_0 | \boldsymbol{X}_{t'}^{1:L}, \boldsymbol{y}), \quad \boldsymbol{X}_{t'}^{1:L} \sim p(\boldsymbol{X}_{t'}^{1:L} | \boldsymbol{X}_0^{1:L}). \tag{5}$$

Then, the corrected $\tilde{\boldsymbol{X}}_0^{1:L}$ can be re-encoded back to resume sampling with an ARM

$$\mathbf{v}^j \sim p_\phi(\mathbf{v}^j | [\tilde{\mathbf{v}}^{1:L}, \mathbf{v}^{L+1:j-1}], \mathbf{u}) \quad \tilde{\mathbf{v}}^{1:L} = \mathcal{E}(\tilde{\boldsymbol{X}}_0^{1:L}). \tag{6}$$

Unlike standard image diffusion models (Rombach et al., 2022), which are limited in their ability to model temporal dynamics, video diffusion models provide a more robust mechanism for ensuring temporal coherence throughout the video. When utilizing diffusion correction, selecting the time step $t' \in [0.4, 0.6]$ allows for control over which aspects of the video are prioritized during refinement. Since global temporal motion is largely established during earlier time steps, higher values such as $t' = 0.6$ are effective for refining temporal motion, while lower values like $t' = 0.4$ are more suitable for addressing local visual artifacts. This approach ensures that both temporal consistency and visual quality are maintained throughout the video generation process.

## 4 EXPERIMENTS

In this section, we test our hypothesis on two distinct experiments: story generation and autoregressive video generation, by applying ACDC to the sampling steps. For all diffusion correction, we employ DDIM sampling (Song et al., 2021a) with classifier free guidance (CFG) (Ho & Salimans, 2021) scale set to 7.5. For the hyperparameters of the base ARM sampling, we use the default settings advised in the original work (Liu et al., 2024; Xie et al., 2024; Lu et al., 2024), unless specified otherwise.

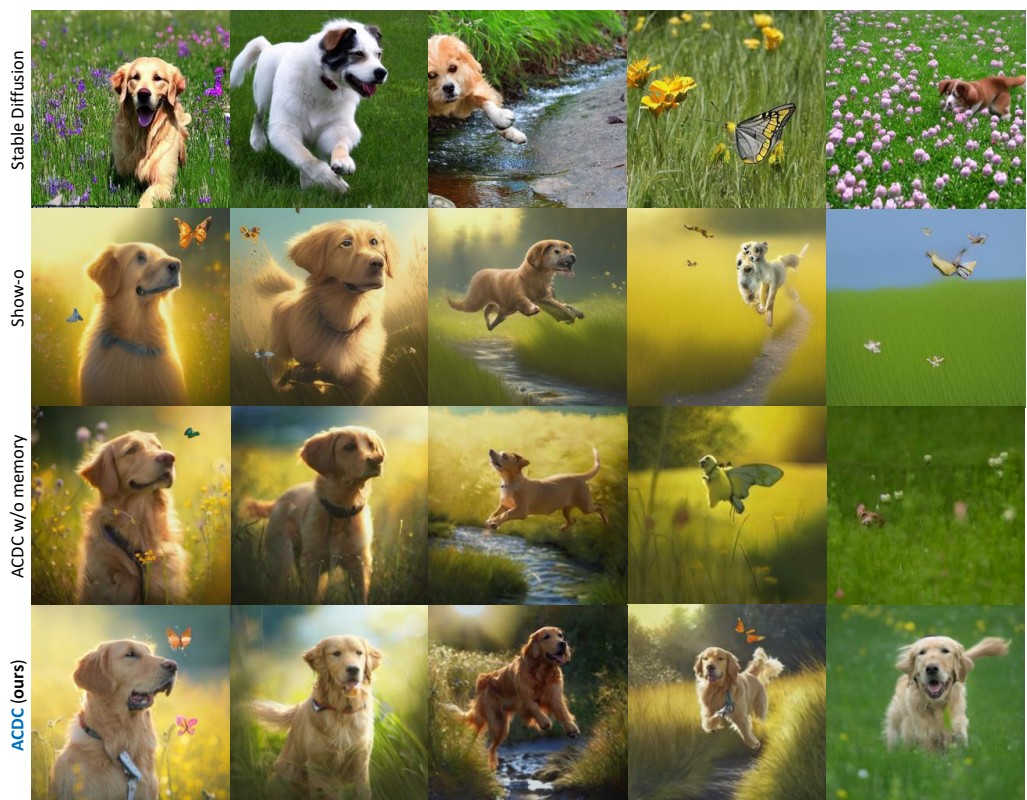

Figure 4: Qualitative comparison of the story generation task. "Golden retriever in sunlit meadow with butterflies (frame 1) ... jumped (frame 3) ... chasing butterfly (frame 4)"

## 4.1 STORY GENERATION

**Benchmark dataset generation** The goal of this section is to study whether ACDC can truly reduce error accumulation and improve the coherent story generation quality. As the base ARM, we choose UIO-2$_{XXL}$ (Lu et al., 2024) and Show-o (Xie et al., 2024), as they are two representative open-sourced models. As the base DM, we choose Stable Diffusion v1.5 (Rombach et al., 2022). We note that other choices such as DeepFloyd IF for DM can also be utilized, without any modifications to the algorithm. To test our hypothesis, it is crucial to select the set of examples that are approximately in-distributed to both the base ARM and DM. We found that existing open-sourced benchmarks such as ChangeIT (Souček et al., 2022) does not meet this criterion, and hence, we decided to generate 1k stories with an LLM. Following the practices in Alpaca (Taori et al., 2023) and self-instruct (Wang et al., 2022), we manually construct 10 examples of the stories and randomly select 3 examples when querying `gpt-4o-mini` to generate another synthetic data. Our stories consist of 6 consecutive prompts, with the first prompt describing the name and the species of the main character and the background it is surrounded in. The following prompts mainly consist of the change in the character's motion and background but is constructed such that a single prompt does not fully describe the context. Further details and examples are provided in Appendix C. The full prompts used to generate the images and videos in this work are gathered in Appendix F.

**Evaluation on the benchmark** We compare the results of applying ACDC to two baseline ARMs, using 4 different metrics: frame consistency (Esser et al., 2023), CLIP similarity, ImageReward (Xu et al., 2024), and FID. Frame consistency is computed as the sum of cosine similarities in the consecutive image frames to capture the consistency of the story. The other three metrics measure the quality of the generated images, as well as the faithfulness to the given text prompt frame-wise. To compute FID against in-distribution images, we generate pseudo-ground truth images using SDXL-lightning (Lin et al., 2024) with 6 NFE.

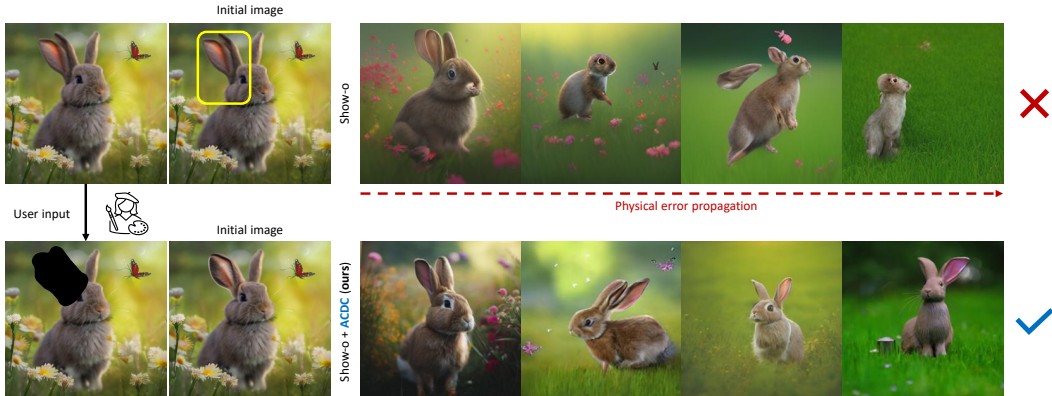

Figure 5: Incorporating user constraints to correct for physical errors in the generated image frames.

In Tab. 1, we observe that ACDC improves the baseline ARMs by significant margins in both frame consistency and image quality metrics. Although the FID score lags behind SD v1.5, this can be partly attributed to the fact that we consider SDXL-generated images as ground truth. In Fig. 1 and Fig. 4, we see that ACDC is capable of generating coherent content, with the same main character with changing background and motion. In contrast, Show-o generates images with artifacts starting from the first frame, which quickly exacerbates as the error accumulates in multiple frame generation. Throughout the generation results, we observe that the generation results from Show-o degrade significantly after the fourth frame. SD generates high quality images frames, but is incapable of understanding the global context and generating a consistent story.

**Correcting physical errors through user constraints**  During the generation process of the base ARM, we often observe physically incorrect images, as illustrated in Fig. 5. Using standard ARM inference, we observe that for such cases, the error accumulates faster, where the generated frames with the same physical error that propagated from the previous frame. In contrast, we observe that after the correction with SD inpainting, the physical error no longer persists, and one can prevent the propagation of physical errors. More examples can be seen in Fig. 9.

**Ablation studies**  On top of SDEdit correction, we have another crucial component of ACDC: LLM memory. We investigate the effect of the memory component in Fig. 4 and Tab. 2. From Fig. 4, we observe that ACDC without memory is as effective as ACDC in the earlier frames, but the performance degrades in the later frames, where without the global context, it is hard to correct for the errors from the ARM, as can be seen in the prompt example in Fig. 3. In Tab. 2, we again see the significant gap of ACDC with and without the memory component.

| Components | | Metrics | | |
|---|---|---|---|---|
| ACDC # | LLM memory | F.Con. (↑) | CLIP ↑ | FID ↓ |
| 0 | ✗ | 0.822 | 28.60 | 112.5 |
| 2 | ✓ | **0.888** | 28.58 | 109.6 |
| 4 | ✓ | 0.843 | 28.79 | 107.6 |
| 6 | ✓ | 0.853 | **30.85** | **101.4** |
| 6 | ✗ | 0.835 | 29.43 | 101.6 |

Table 2: Ablation study in story generation task with 100 test stories.

Further, we study the effect of ACDC by controlling the number of frames that the correction is applied. ACDC # 2 refers to the case where we apply the correction to the first two frames and the following frames are generated only through ARM. Two facts are notable: 1) ACDC enhances the result even when we do not correct for all the frames, showing that one can reduce the exponential error accumulation, 2) Correcting for all the frames performs the best, although the frame consistency is slightly compromised compared to ACDC # 2.

### 4.2 AUTOREGRESSIVE VIDEO GENERATION

**Experimental settings**  We conducted experiments using the Large World Model (LWM) (Liu et al., 2024)—a large multimodal model capable of generating videos frame-by-frame—to assess the impact of ACDC on reducing error propagation in video generation task. We observed that LWM

| Method | Subject consistency (↑) | Background Consistency (↑) | Motion Smoothness (↑) | Dynamic Degree (↑) | Aesthetic Quality (↑) | Imaging Quality (↑) |
|---|---|---|---|---|---|---|
| LWM (Liu et al., 2024) | 0.7369 | 0.8695 | 0.9479 | 0.7018 | 0.4105 | 0.5127 |
| LWM (Liu et al., 2024) + ACDC (ours) | **0.7622**▲3.43% | **0.8821**▲1.45% | **0.9501**▲0.23% | **0.7099**▲1.15% | **0.4406**▲7.33% | **0.5164**▲0.72% |

Table 3: Quantitative evaluation of the autoregressive video generation task.

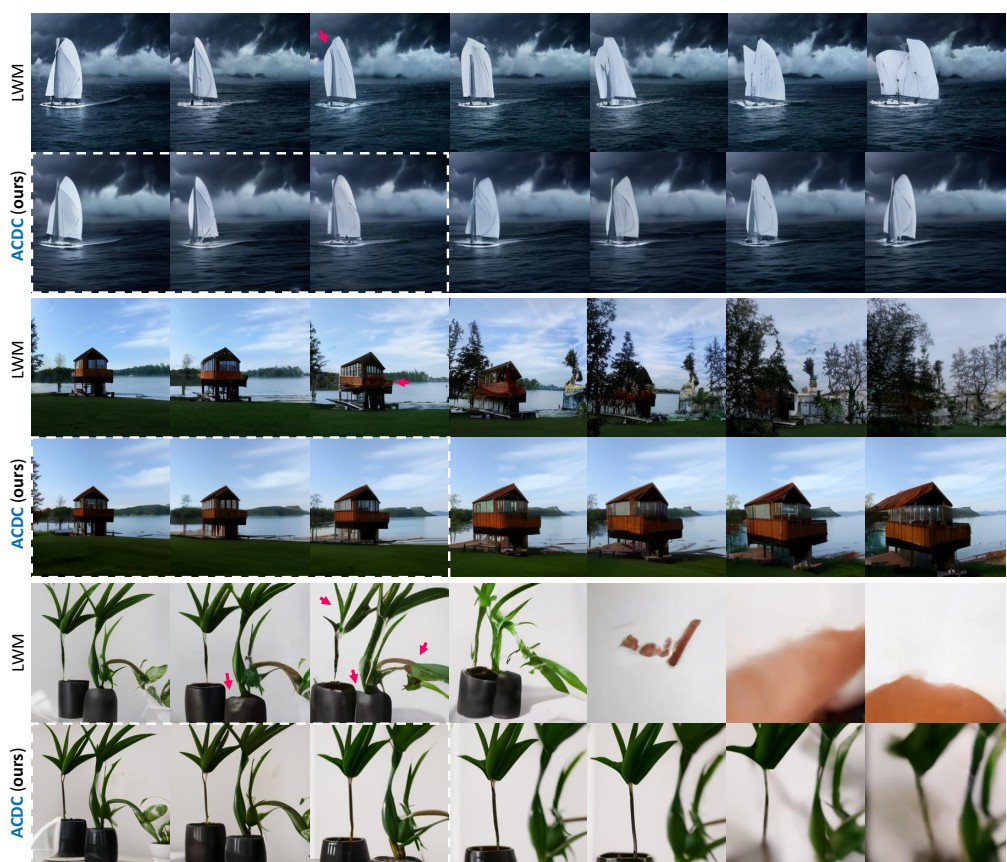

Figure 6: Qualitative comparison of the autoregressive video generation task. White dotted box refers to the first 16 frames, which are corrected by ACDC. The remaining frames are left uncorrected. Magenta arrows indicate regions where visual artifacts or inconsistencies with preceding frames occur in the original video.

frequently exhibits visual artifacts and content inconsistency when generating sequences longer than 16 frames. To investigate the effectiveness of ACDC, we compared two generation pipelines: (1) directly generating all $N = 32$ frames using LWM and (2) generating the first $L = 16$ frames with LWM, applying ACDC using AnimateDiff (Guo et al., 2024) to correct the frames, and subsequently generating the remaining 16 frames. For both pipelines, we introduced diversity in the initial frame by setting the CFG scale to $5.0$ and utilizing top-$k$ sampling with $k = 8192$. The subsequent frames were generated with a reduced CFG scale of $1.0$ and top-$k$ sampling with $k = 1000$ to ensure consistency. All experiments were performed on 4 NVIDIA H100 GPUs. These experiments allowed us to evaluate the role of ACDC in mitigating error propagation and preserving video quality across extended sequences.

**Dataset and Evaluation on the benchmark** To evaluate the video generation and correction capabilities across a diverse range of subjects and motions, we utilized 800 prompts from VBench (Huang et al., 2024b), spanning multiple categories. In addition, we assessed both the spatial and temporal quality of the generated videos using 2 spatial and 4 temporal metrics, leveraging the tools provided

by VBench for accurate measurement. For spatial quality, we employed the LAION aesthetic predictor (LAION-AI, 2022) to assess frame-wise aesthetic quality, and the Multi-Scale Image Quality Transformer (MUSIQ) (Ke et al., 2021) to quantify imaging quality, specifically evaluating noise and blur levels in each frame. For temporal quality assessment, we measured subject consistency and background consistency between frames using DINO (Caron et al., 2021) feature similarity and CLIP similarity, respectively. To mitigate the potential risk of improving consistency at the expense of dynamic motion, we employed RAFT (Teed & Deng, 2020) to measure the dynamic degree and verify whether the observed consistency enhancements were achieved without hindering the dynamic quality of motion. Lastly, to assess the smoothness of generated motion, we compared the generated motion against the predicted motion from a video frame interpolation model (Li et al., 2023b), allowing us to quantify the smoothness of motion transitions between frames.

In Tab. 3, we observed that ACDC improves both subject and background consistency compared to the baseline LWM, with a notably large margin. Importantly, the dynamic degree did not decrease, indicating that the improvement in consistency was achieved without the video becoming overly static. Furthermore, both imaging quality and aesthetic quality showed improvements, with aesthetic quality exhibiting a substantial increase, likely driven by the influence of the video diffusion model. As demonstrated in Fig. 1 and Fig. 6, LWM exhibits minor visual artifacts and content inconsistencies in the initial 16 frames (first 3 figures in each row), which, if left uncorrected, progressively worsen in subsequent frames, resulting in more pronounced artifacts and content drift. However, by applying ACDC to correct the first 16 frames, these issues are effectively mitigated, enabling the consistency in later frames to remain aligned with the corrected early frames, thus preventing the propagation of errors throughout the sequence.

## 5    CONCLUSION

We presented Autoregressive Coherent multimodal generation with Diffusion Correction (ACDC), a flexible, zero-shot approach that leverages the complementary strengths of autoregressive models (ARMs) and diffusion models (DMs) for multimodal generation tasks. By employing DMs as local correctors through SDEdit and incorporating a memory module with large language models (LLMs), ACDC addresses the exponential error accumulation inherent in ARMs, resulting in consistent and high-quality outputs across a range of tasks. Our experiments on coherent story generation and autoregressive video generation validate that ACDC effectively integrates pre-trained ARMs and DMs without requiring additional fine-tuning or architectural modifications, thus establishing a robust and scalable framework for multimodal generation. Future work could explore further enhancements in the memory module and investigate the application of ACDC to other complex multimodal tasks, pushing the boundaries of generative modeling.

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

# A   OVERVIEW OF DIFFUSION MODELS

In diffusion models (Ho et al., 2020; Song et al., 2021c), we first define the forward noising process with a stochastic differential equation (SDE). The variance preserving (VP) forward diffusion trajectory is given by the following SDE (Song et al., 2021c; Ho et al., 2020)

$$d\boldsymbol{x} = -\frac{1}{2}\beta(t)\boldsymbol{x}dt + \sqrt{\beta(t)}d\mathbf{w}, \tag{7}$$

with $\beta(t) = \beta_{min} + (\beta_{max} - \beta_{min})t$, $\boldsymbol{x}(t) \in \mathbb{R}^d$, and $t \in [0, 1]$. Here, $\mathbf{w}$ is a standard $d-$dimensional Wiener process. An intriguing property of the forward diffusion is that the forward perturbation kernel is given in a closed form

$$p(\boldsymbol{x}(t)|\boldsymbol{x}(0)) = \mathcal{N}(\boldsymbol{x}(t); \sqrt{\bar{\alpha}(t)}\boldsymbol{x}(0), (1 - \bar{\alpha}(t))\boldsymbol{I}), \tag{8}$$

where $\bar{\alpha}(t) = \exp\left(\int_0^t \frac{1}{2}\beta(u)\,du\right)$. Here, it is important to note that the parameters $\beta_{min}$ and $\beta_{max}$ are chosen such that $\bar{\alpha}(0) \approx 1$ and $\bar{\alpha}(1) \approx 0$ such that as the forward diffusion approaches the terminal state $t \to 1$, $p_1(\boldsymbol{x})$ approximates standard normal distribution. The reverse SDE of Eq. (7) reads

$$d\boldsymbol{x} = \left[-\frac{1}{2}\beta(t)\boldsymbol{x} - \beta(t)\nabla_{\boldsymbol{x}} \log p_t(\boldsymbol{x})\right] dt + \sqrt{\beta(t)}\mathbf{w}. \tag{9}$$

Sampling data from noise $\boldsymbol{x} \sim p_{data}(\boldsymbol{x})$ amounts to solving Eq. (9) from $t = 1$ to $t = 0$, which is governed by the score function $\nabla_{\boldsymbol{x}} \log p_t(\boldsymbol{x})$. The score function is trained with the denoising score matching (DSM) objective

$$\min_{\boldsymbol{\theta}} \mathbb{E}_{t, \boldsymbol{x}(t) \sim p(\boldsymbol{x}(t)|\boldsymbol{x}(0)), \boldsymbol{x}(0) \sim p(\boldsymbol{x})} \left[\lambda(t)\|s_{\boldsymbol{\theta}}(\boldsymbol{x}(t), t) - \nabla_{\boldsymbol{x}_t} \log p(\boldsymbol{x}(t)|\boldsymbol{x}(0))\|_2^2\right], \tag{10}$$

where $\lambda(t)$ is a weighting function. The score function can be shown to be equivalent to denoising autoencoders (Karras et al., 2022) through Tweedie's formula (Efron, 2011), and also equivalent to epsilon matching (Ho et al., 2020). When plugging in the trained score function to Eq. (9), we have the empirical reverse SDE

$$d\boldsymbol{x} = \left[-\frac{1}{2}\beta(t)\boldsymbol{x} - \beta(t)s_{\boldsymbol{\theta}}(\boldsymbol{x}(t), t)\right] dt + \sqrt{\beta(t)}\mathbf{w}. \tag{11}$$

Running Eq. (11) will sample from $\boldsymbol{x} \sim p_\theta(\boldsymbol{x})$, with $p_\theta(\boldsymbol{x}) = p_{data}(\boldsymbol{x})$ if $\nabla_{\boldsymbol{x}} \log p_t(\boldsymbol{x}) \equiv s_{\boldsymbol{\theta}}(\boldsymbol{x}(t), t)$ for all $t \in [0, 1]$ (Song et al., 2021b). It is notable that the time-marginal distributions of $Eq.$ (9) can also be retrieved by running the so-called probability-flow ODE (PF-ODE)

$$d\boldsymbol{x} = -\frac{1}{2}\beta(t) \left[\boldsymbol{x} + \nabla_{\boldsymbol{x}} \log p_t(\boldsymbol{x})\right] dt. \tag{12}$$

We can again use a plug-in estimate for the score term, achieving the empirical PF-ODE

$$d\boldsymbol{x} = -\frac{1}{2}\beta(t) \left[\boldsymbol{x} + s_{\boldsymbol{\theta}}(\boldsymbol{x}, t)\right] dt. \tag{13}$$

Modern foundational diffusion models are typically text-conditioned (Rombach et al., 2022; Team, 2023), so that the user can control the content to be generated. Denoting $\boldsymbol{y}$ as the conditioning text, the task of text-to-image or text-to-video generation can be considered sampling from the posterior $p(\boldsymbol{x}|\boldsymbol{y})$. In order to do this, we train a conditional score function with *paired* (image, text) data

$$\min_{\boldsymbol{\theta}} \mathbb{E}_{t, \boldsymbol{x}(t) \sim p(\boldsymbol{x}(t)|\boldsymbol{x}(0)), (\boldsymbol{x}(0), \boldsymbol{y}) \sim p(\boldsymbol{x}, \boldsymbol{y})} \left[\lambda(t)\|s_{\boldsymbol{\theta}}(\boldsymbol{x}(t), \boldsymbol{y}, t) - \nabla_{\boldsymbol{x}_t} \log p(\boldsymbol{x}(t)|\boldsymbol{x}(0))\|_2^2\right], \tag{14}$$

where we denote the empirical joint distribution as $p(\boldsymbol{x}, \boldsymbol{y})$. In the optimal case, we hae $\nabla_{\boldsymbol{x}} \log p_t(\boldsymbol{x}|\boldsymbol{y}) \equiv s_{\boldsymbol{\theta}}(\boldsymbol{x}(t), \boldsymbol{y}, t)$. Sampling from the posterior can be achieved by plugging in the conditional score function to Eq. (9) or Eq. (12).

## B  THEORETICAL JUSTIFICATION

Our diffusion correction algorithm starts from an initial point $\boldsymbol{x}$, which was obtained by detokenizing the sampled output of an ARM. We first perturb it with forward diffusion

$$\boldsymbol{x}(t') = \sqrt{\bar{\alpha}(t')}\boldsymbol{x} + \sqrt{1 - \bar{\alpha}(t')}\boldsymbol{\epsilon}, \quad \boldsymbol{\epsilon} \sim \mathcal{N}(0, \boldsymbol{I}), \tag{15}$$

then recovers it through the conditional reverse PF-ODE from time $t'$ to 0

$$\tilde{\boldsymbol{x}}(0) = \int_{t'}^{0} -\frac{1}{2}\beta(t)[\boldsymbol{x} + s_{\boldsymbol{\theta}}(\boldsymbol{x}, \boldsymbol{y}, t)]dt, \tag{16}$$

where $\boldsymbol{y}$ is the text condition corrected through our LLM memory module. Let $p_t(\boldsymbol{x})$ the distribution of forward-diffused clean data obtained through Eq. (7), and $q_t(\boldsymbol{x})$ be the distribution of the forward-diffused corrupted data from the ARM sampling, again obtained through Eq. (7).

**Theorem 1** (Nie et al. (2022)). *The KL divergence between $p_t$ and $q_t$ monotonically decreases through forward diffusion, i.e.*

$$\frac{\partial D_{KL}(p_t || q_t)}{\partial t} \leq 0. \tag{17}$$

Adding Gaussian noise to a random variable as in Eq. (7) leads to convolution with a Gaussian blur kernel for distributions, and hence $p_t, q_t$ are blurred distributions. Theorem 1 states by blurring the distribution, the two become close together, and there must exist $t' \in [0, 1]$ with $D_{KL}(p_{t'} || q_{t'}) < \varepsilon$. Recovering with Eq. (16) will guarantee that we pull the sample closer toward the desired distribution.

However, care must be taken since we should also keep the corrected image close to the starting point. To see if this hold, we start from a simplified case without considering the text condition (i.e. null conditioning). Let us define $\hat{\boldsymbol{x}}(0)$ as follows

$$\hat{\boldsymbol{x}}(0) = \int_{t'}^{0} -\frac{1}{2}\beta(t)[\boldsymbol{x} + s_{\boldsymbol{\theta}}(\boldsymbol{x}, t)]dt. \tag{18}$$

We have the following result

**Theorem 2.** *Assume that the score function is globally bounded by $\|s_{\boldsymbol{\theta}}(\boldsymbol{x}, t)\| \leq C$. Then,*

$$\mathbb{E}[\|\hat{\boldsymbol{x}}(0) - \boldsymbol{x}\|] \leq (1 - \sqrt{\bar{\alpha}(t')}) \left( \eta + \frac{C}{\bar{\alpha}(t')} \right) + \frac{\sqrt{1 - \bar{\alpha}(t')}\sqrt{d}}{\sqrt{\bar{\alpha}(t')}} \tag{19}$$

*where $\eta = \mathbb{E}[\|\boldsymbol{x}\|]$.*

*Proof.* By reparametrization, the forward diffusion to $t'$ in Eq. (7) reads

$$\boldsymbol{x}(t') = \sqrt{\bar{\alpha}(t')}\boldsymbol{x} + \sqrt{1 - \bar{\alpha}(t')}\boldsymbol{\epsilon}, \quad \boldsymbol{\epsilon} \sim \mathcal{N}(0, \boldsymbol{I}). \tag{20}$$

Let $\mu(t)$ be the integrating factor

$$\mu(t) = \exp\left( \int_{t'}^{t} \frac{1}{2}\beta(u)\, du, \right) \tag{21}$$

where by definition, $\mu(t') = 1$ and $\mu(0) = 1/\sqrt{\bar{\alpha}(t')}$. Multiplying $\mu(t)$ on both sides of Eq. (18),

$$\frac{d}{dt}(\mu(t)\boldsymbol{x}(t)) = -\frac{1}{2}\beta(t)\mu(t)s_{\boldsymbol{\theta}}(\boldsymbol{x}, t). \tag{22}$$

Integrating both sides from $t'$ to 0,

$$\mu(0)\hat{\boldsymbol{x}}(0) - \mu(t')\boldsymbol{x}(t') = -\frac{1}{2}\int_{t'}^{0} \beta(t)\mu(t)s_{\boldsymbol{\theta}}(\boldsymbol{x}, t)dt. \tag{23}$$

Rearranging,

$$\hat{\boldsymbol{x}}(0) = \frac{1}{\mu(0)}\boldsymbol{x}(t') - \frac{1}{2\mu(0)}\int_{t'}^0 \beta(u)\mu(u)s_{\boldsymbol{\theta}}(\boldsymbol{x}(u), u)\, du \tag{24}$$

$$= \sqrt{\bar{\alpha}(t')}\boldsymbol{x} + \frac{\sqrt{1 - \bar{\alpha}(t')}}{\sqrt{\bar{\alpha}(t')}}\boldsymbol{\epsilon} - \frac{1}{2\sqrt{\bar{\alpha}(t')}}\int_{t'}^0 \beta(u)\mu(u)s_{\boldsymbol{\theta}}(\boldsymbol{x}(u), u)\, du \tag{25}$$

Hence,

$$\hat{\boldsymbol{x}}(0) - \boldsymbol{x} = (\sqrt{\bar{\alpha}(t')} - 1)\boldsymbol{x} + \frac{\sqrt{1 - \bar{\alpha}(t')}}{\sqrt{\bar{\alpha}(t')}}\boldsymbol{\epsilon} - \frac{1}{2\sqrt{\bar{\alpha}(t')}}\int_{t'}^0 \beta(u)\mu(u)s_{\boldsymbol{\theta}}(\boldsymbol{x}(u), u)\, du. \tag{26}$$

Taking expectation on both sides and leveraging triangle inequality, we have

$$\mathbb{E}[\|\hat{\boldsymbol{x}}(0) - \boldsymbol{x}\|] \leq \underbrace{\mathbb{E}[\|(1 - \sqrt{\bar{\alpha}(t')})\boldsymbol{x}\|]}_{T_1} \tag{27}$$

$$+ \underbrace{\mathbb{E}\left[\left\|\frac{\sqrt{1 - \bar{\alpha}(t')}}{\sqrt{\bar{\alpha}(t')}}\boldsymbol{\epsilon}\right\|\right]}_{T_2} \tag{28}$$

$$+ \underbrace{\mathbb{E}\left[\frac{1}{2\sqrt{\bar{\alpha}(t')}}\left\|\int_{t'}^0 \beta(u)\mu(u)s_{\boldsymbol{\theta}}(\boldsymbol{x}(u), u)\, du\right\|\right]}_{T_3} \tag{29}$$

$T_1 = (1 - \sqrt{\bar{\alpha}(t')})\mathbb{E}[\|\boldsymbol{x}\|]$. From $\mathbb{E}[\|\boldsymbol{\epsilon}\|] \leq \sqrt{d}$, $T_2 \leq \frac{\sqrt{1 - \bar{\alpha}(t')}\sqrt{d}}{\sqrt{\bar{\alpha}(t')}}$. Finally, from the boundedness assumption of $s_{\boldsymbol{\theta}}(\boldsymbol{x}, t)$,

$$\left\|\int_{t'}^0 \beta(u)\mu(u)s_{\boldsymbol{\theta}}(\boldsymbol{x}(u), u)\, du\right\| \leq C\int_{t'}^0 \beta(u)\mu(u)\, du \tag{30}$$

$$= 2C\int_{t'}^0 \frac{d\mu(u)}{du}du \tag{31}$$

$$= 2C(\mu(0) - \mu(t')) \tag{32}$$

$$= 2C\left(\frac{1 - \sqrt{\bar{\alpha}(t')}}{\sqrt{\bar{\alpha}(t')}}\right), \tag{33}$$

so $T_3 = C\frac{1 - \sqrt{\bar{\alpha}(t')}}{\bar{\alpha}(t')}$. Summing up,

$$\mathbb{E}[\|\hat{\boldsymbol{x}}(0) - \boldsymbol{x}\|] \leq (1 - \sqrt{\bar{\alpha}(t')})\left(\eta + \frac{C}{\bar{\alpha}(t')}\right) + \frac{\sqrt{1 - \bar{\alpha}(t')}\sqrt{d}}{\sqrt{\bar{\alpha}(t')}} \tag{34}$$

$\square$

Recall that $\beta(t)$ is designed such that $\bar{\alpha}(0) \approx 1$ and $\bar{\alpha}(1) \approx 0$. Hence, Theorem 2 implies that the deviation from $\boldsymbol{x}$ after SDEdit, if $t'$ is not too large, is bounded. Next, we further consider the case of $\tilde{\boldsymbol{x}}(0)$ in Eq. (16)

**Theorem 3.** *Assume that $\|s_{\boldsymbol{\theta}}(\boldsymbol{x}_t, \boldsymbol{y}, t) - s_{\boldsymbol{\theta}}(\boldsymbol{x}_t, \tilde{\boldsymbol{y}}, t)\| \leq Kd(\boldsymbol{y}, \tilde{\boldsymbol{y}})$, where $d(\cdot, \cdot)$ measures the feature distance between the text conditions $\boldsymbol{y}$ and $\tilde{\boldsymbol{y}}$. Let $\tilde{\boldsymbol{x}}(0)$ be the SDEdit reconstruction using time $t'$ and the conditional score $s_{\boldsymbol{\theta}}(\boldsymbol{x}_t, \boldsymbol{y}, t)$ for solving the PF-ODE. Then,*

$$\mathbb{E}[\|\tilde{\boldsymbol{x}}(0) - \boldsymbol{x}\|] \leq (1 - \sqrt{\bar{\alpha}(t')})\left(\eta + \frac{C}{\bar{\alpha}(t')} + K\bar{\alpha}(t')d(\boldsymbol{y}, \tilde{\boldsymbol{y}})\right) + \frac{\sqrt{1 - \bar{\alpha}(t')}\sqrt{d}}{\sqrt{\bar{\alpha}(t')}}. \tag{35}$$

*Proof.*

$$\|\tilde{\boldsymbol{x}}(0) - \boldsymbol{x}\| = \|\tilde{\boldsymbol{x}}(0) - \hat{\boldsymbol{x}}(0) + \hat{\boldsymbol{x}}(0) - \boldsymbol{x}\| \tag{36}$$

$$\leq \|\tilde{\boldsymbol{x}}(0) - \hat{\boldsymbol{x}}(0)\| + \|\hat{\boldsymbol{x}}(0) - \boldsymbol{x}\| \tag{37}$$

by triangle inequality. The second term in the RHS is given by Theorem 2. Regarding the first term of the RHS,

$$\|\tilde{\boldsymbol{x}}(0) - \hat{\boldsymbol{x}}(0)\| \leq \frac{K d(\boldsymbol{y}, \tilde{\boldsymbol{y}})}{2\mu(0)} \int_{t'}^{0} \beta(u)\mu(u) \, du \tag{38}$$

$$= K \frac{1 - \sqrt{\bar{\alpha}(t')}}{\bar{\alpha}(t')} d(\boldsymbol{y}, \tilde{\boldsymbol{y}}). \tag{39}$$

Thus,

$$\|\tilde{\boldsymbol{x}}(0) - \boldsymbol{x}\| = \leq (1 - \sqrt{\bar{\alpha}(t')}) \left( \eta + \frac{C}{\bar{\alpha}(t')} + K\bar{\alpha}(t')d(\boldsymbol{y}, \tilde{\boldsymbol{y}}) \right) + \frac{\sqrt{1 - \bar{\alpha}(t')}\sqrt{d}}{\sqrt{\bar{\alpha}(t')}}. \tag{40}$$

$\square$

Theorem 3 shows that for the conditional case, in order to control the deviation from $\boldsymbol{x}$, an additional factor of the *correctness* of the text prompt should be considered. Only when $\boldsymbol{y} = \tilde{\boldsymbol{y}}$, the bound matches the unconditional case. Otherwise, there is larger deviation. Note that due to the discrepancy between how the text conditioning is used in ARMs and DMs, we do not know the underlying "ground truth" $\boldsymbol{y}$. Hence, Theroem 3 implies that it is important that we use the memory module to *align* the text condition used in the SDEdit process using the memory module.

---

**System:**
You are a helpful and creative story generator. You should brainstorm creative and impressive yet concise stories so that a text-to-image generative AI will be able to easily generate the images with it. You are given 3 such example stories. Reference examples are delimited with """ as a guide. Each story will consist of 6 sentences separated with linebreaks.
**User:**
Similar to the examples above, a single story consists of 6 sentences. The story is about an animal main character with a name. The prompts should be simple and concise. The first sentence should describe the background. The following prompts should describe the motion of the main character. Be creative, and do not make a copy of the above examples.
Generated story:

---

Table 4: System, user prompt used to generate new stories.

---

**Example 1:**
1. Max, the corgi, stood in a sunny park with tall trees, a blue sky, and children playing in the distance.
2. Max stopped and sniffed the ground, looking curious.
3. He barked excitedly, watching a butterfly flutter nearby.
4. The corgi jumped up, trying to catch the butterfly in mid-air.
5. Max spun around playfully in circles, chasing his tail.
6. Max lay down panting, tongue out and happy.
**Example 2:**
1. A rabbit was in a quiet meadow surrounded by wildflowers, with a gentle stream flowing nearby.
2. Thumper, the rabbit, stood on its hind legs, sniffing the air in the meadow.
3. Thumper hopped forward cautiously, ears twitching.
4. The rabbit nibbled on a small patch of grass, looking relaxed.
5. She perked up and looked around, alert for any sounds.
6. She dashed forward, making a quick zig-zag in the field.

---

Table 5: Example stories that is fed to GPT-4o to generate new stories.

> **Generated 1:**
> 1. Benny the beagle sat in a vibrant garden filled with blooming flowers and buzzing bees under a clear blue sky.
> 2. His nose twitched as he picked up the sweet scent of roses and lavender.
> 3. Benny jumped up and trotted toward the colorful flowerbeds, tail wagging with excitement.
> 4. He playfully pawed at the petals, sending a few fluttering to the ground.
> 5. The beagle barked joyfully at a bumblebee that dared to dance close to his snout.
> 6. Finally, Benny sprawled out on the soft grass, basking in the sun with a contented sigh.
>
> **Generated 2:**
> 1. Oliver the owl perched high on an ancient oak tree, the moonlight casting a silvery glow over the tranquil meadow below.
> 2. He spread his wings wide, feeling the night air beneath him, and took off into the starry sky.
> 3. With a graceful glide, Oliver swooped down, searching for the shimmering fireflies dancing in the moonlight.
> 4. The wise owl twirled mid-air, playfully chasing after a particularly bright firefly.
> 5. He landed softly on a fence post, watching as the glowing lights flickered around him.
> 6. Oliver hooted joyfully, reveling in the magic of the night and the beauty of his surroundings.

Table 6: Generated stories with GPT-4o.

---

**Algorithm 1** GPT-4o Story Generation with Random Examples.

---

**Input:** `system_prompt`, `user_prompt`, N examples .
  client = OpenAI()
**Output:** Generated `story`.
  `examples` = ""
  random_integers = random.sample(range(1, N), 3)
  **for** random_integer **in** random_integers **do**
    `file_path` = example_prompt_directory + random_integer:04d + ".txt"
    Open `file_path` and read content into `example`
    Append `example` to `examples` surrounded by formatting
  **end for**
  `system_prompt` += `examples`
  completion = client.chat.completions.create(
    model="gpt-4o",
    temperature=1.0,
    messages=[
      {"role": "system", "content": `system_prompt`},
      {"role": "user", "content": `user_prompt`},
    ]
  )
  `story_string` = completion.choices[0].message["content"]
  `story` = parser(story_string)
  **Return** `story`

---

## C    DATA GENERATION

As stated in Sec. 4, we first manually construct 10 example stories to be fed to the LLM. The examples are shown in Tab. 1. As shown in Tab. 4 and Alg. 1, we aim to sample diverse stories with the LLM by using 1.0 for the temperature, and prompting the model to be creative in its choice. Some examples of the generated stories are presented in Tab. 6.

## D    LLM MEMORY MODULE

We explored two design choices for the LLM memory module. First, one could query the LLM for every image $x_0^{(i)}$ to be generated. Alternatively, one could query the LLM one time with all the story prompts, and ask it to causally refine it. As we did not observe much difference in the quality, we

**System:**
You are a helpful text refiner AI. Given a story consisting of 6 sentences, you will causally modify the second sentence and onwards so that it contains the crucial information previous sentences. The second sentence should contain information sentence 1 and 2, The third sentence should contain information from sentence 1-3, and so on. The refined sentences should infer the referent from the contextual cues and make it explicit. Reference examples are delimited with """ as a guide.
"""
(Original story)
Max, the corgi, stood in a sunny park with tall trees, a blue sky, and children playing in the distance.
Max stopped and sniffed the ground, looking curious.
Max barked excitedly, watching a butterfly flutter nearby.
The corgi jumped up, trying to catch the butterfly in mid-air.
Max spun around playfully in circles, chasing his tail.
Max lay down panting, tongue out and happy.
(Modified story)
Max, the corgi, stood in a sunny park with tall trees, a blue sky, and children playing in the distance.
Max, the corgi, stopped and sniffed the ground, looking curious.
Max, the corgi, barked excitedly, watching a butterfly flutter nearby.
Max, the corgi jumped up, trying to catch the butterfly in mid-air.
Max, the corgi, spun around playfully in circles, chasing his tail.
Max, the corgi, lay down panting, tongue out and happy.
"""
**User:**
Similar to the examples above, modify the sentences:
"""
(Original story)
{original_story}
(Modified story)

Table 7: LLM memory module system and user prompt

decided to choose the latter. The prompt design used for the LLM memory module is provided in Tab. 7. For the LLM, we used `gpt-4o-mini` for our experiments but found open-sourced models such as `gemma-9b-it` (Team et al., 2024) also produced similar results.

# E    FURTHER EXPERIMENTAL RESULTS

# F    PROMPTS FOR RESULTS

**Fig 1**

- **Row 1-2.**
  - Prompt 1: "Luna, the fox, stood at the edge of a moonlit forest vibrant with glowing fireflies."
  - Prompt 2: "With a flick of her bushy tail, she slipped stealthily through the underbrush, her paws barely making a sound."
  - Prompt 3: "Luna darted between the trees, her amber eyes glinting in the dim light as she chased a fluttering moth."
  - Prompt 4: "Suddenly, she stopped in her tracks, her ears perked up at the rustle of leaves nearby."
  - Prompt 5: "With a playful bounce, Luna leaped over a fallen log, determined not to lose her target."
- **Row 3-4:**
  - Prompt 1: "Benny the beagle sat in a vibrant garden filled with blooming flowers and buzzing bees under a clear blue sky."
  - Prompt 2: "His nose twitched as he picked up the sweet scent of roses and lavender."

- **Prompt 3:** "Benny jumped up and trotted toward the colorful flowerbeds, tail wagging with excitement."
  - **Prompt 4:** "He playfully pawed at the petals, sending a few fluttering to the ground."
  - **Prompt 5:** "The beagle barked joyfully at a bumblebee that dared to dance close to his snout."
- **Row 5-6:** "A male interviewer listening to a person talking."

**Fig 4**

- Prompt 1: "Baxter, the adventurous golden retriever, stood at the edge of a sunlit meadow dotted with wildflowers and fluttering butterflies."
- Prompt 2: "With a wagging tail, he bounded forward, the soft grass giving way beneath his paws."
- Prompt 3: "Baxter jumped playfully, leaping over a small brook that shimmered in the sunlight."
- Prompt 4: "He chased after a bright yellow butterfly, his golden coat glistening as he ran."
- Prompt 5: "Baxter skidded to a halt, nose to the ground, investigating the sweet scent of blooming clover."

**Fig 5**

- Prompt 1: "The curious rabbit hopped through a vibrant meadow, wildflowers swaying gently in the summer breeze."
- Prompt 2: "Ruby, the rabbit, bounded playfully between patches of colorful blooms."
- Prompt 3: "With a swift turn, Ruby, the rabbit, darted towards a butterfly fluttering nearby, her ears perked with excitement."
- Prompt 4: "Ruby, the rabbit, leaped gracefully, her paws barely brushing the ground as she chased the delicate creature."
- Prompt 5: "Suddenly, Ruby, the rabbit, skidded to a halt, captivated by a glimmering stream sparkling in the sunlight."

**Fig 6**

- **Row 1-2:** "A boat sailing on a stormy ocean."
- **Row 3-4:** "A wooden house overseeing the lake"
- **Row 5-6:** "An elegant ceramic plant pot and hanging plant on indoor"

**Fig 10**

- **Row 1-2:** "Aerial shot of interior of the warehouse."
- **Row 3-4:** "Aerial video of Stuttgart tv tower in Germany."
- **Row 5-6:** "A chef holding and checking kitchen utensils."
- **Row 7-8:** "A scenic view of the golden hour at Norway."
- **Row 9-10:** "Close up view of lighted candles."

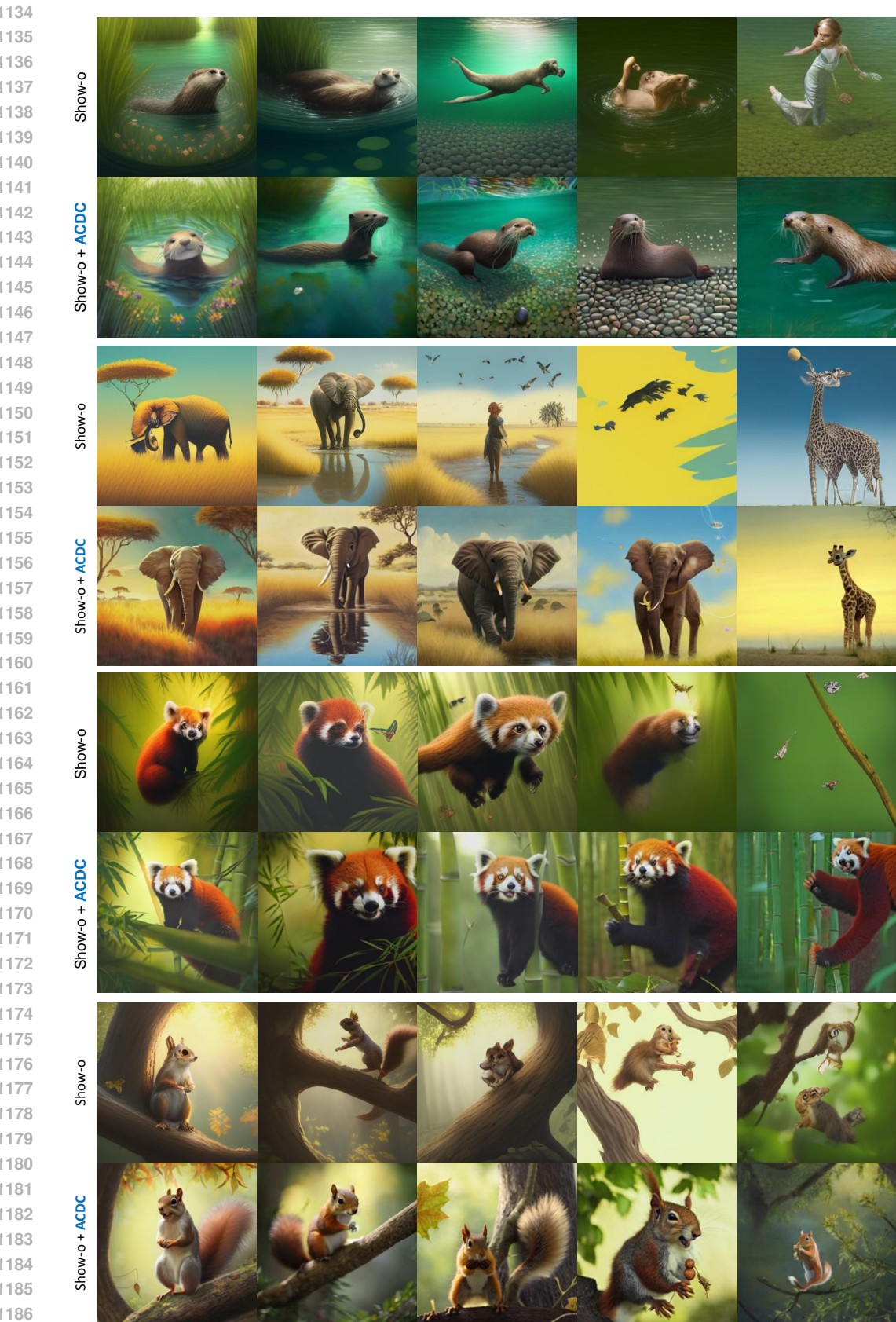

Figure 7: Further results of ACDC applied to Show-o.

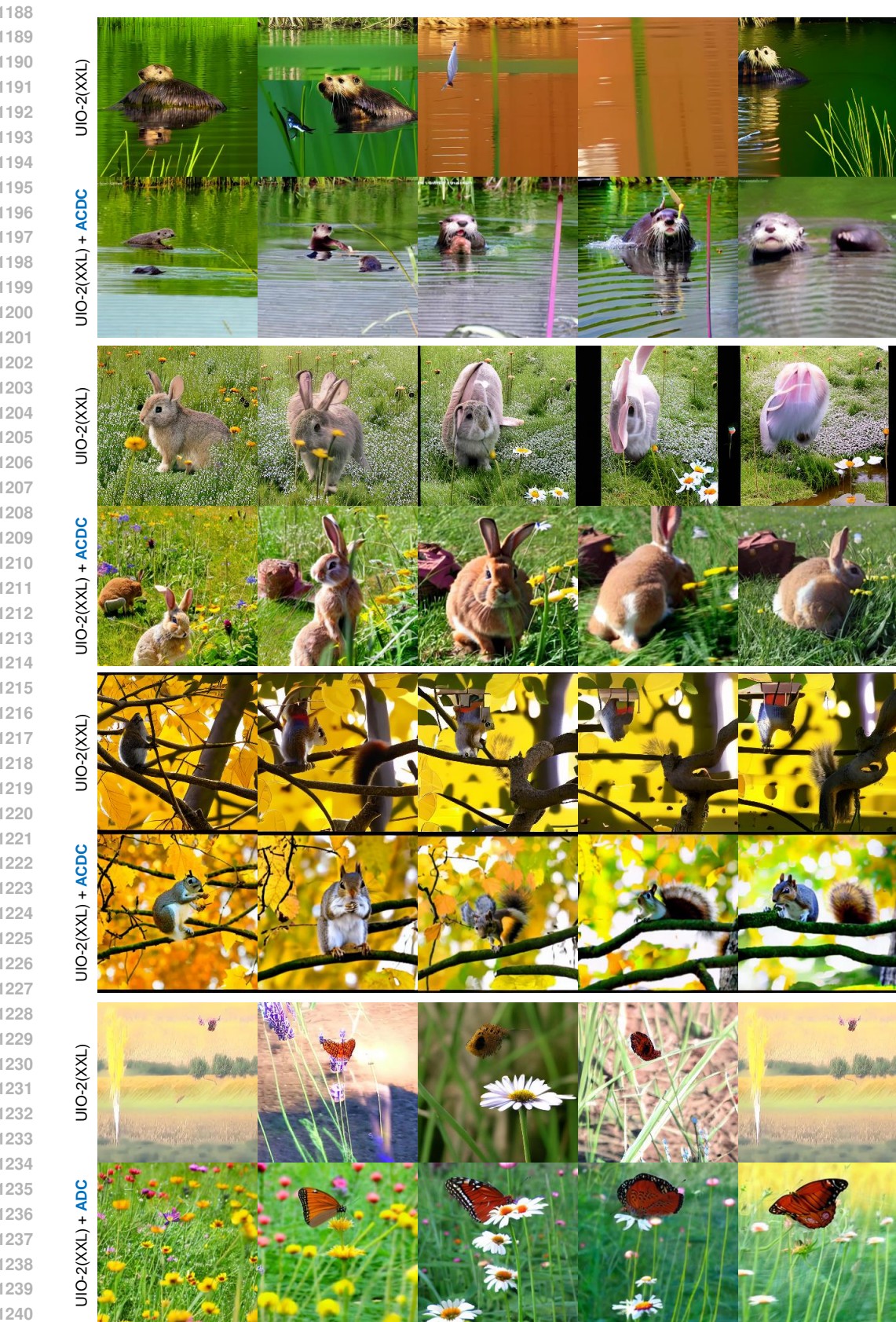

Figure 8: Further results of ACDC applied to Unified-IO-2.

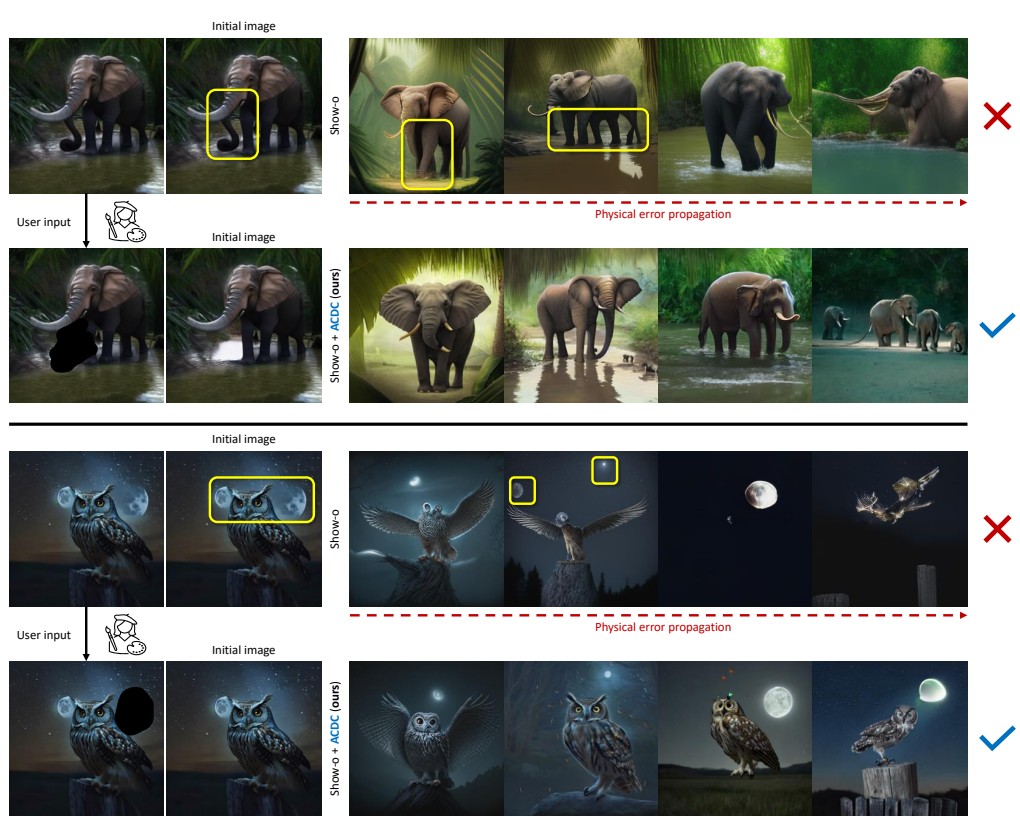

Figure 9: Incorporating user constraints to correct for physical errors in the generated image frames.

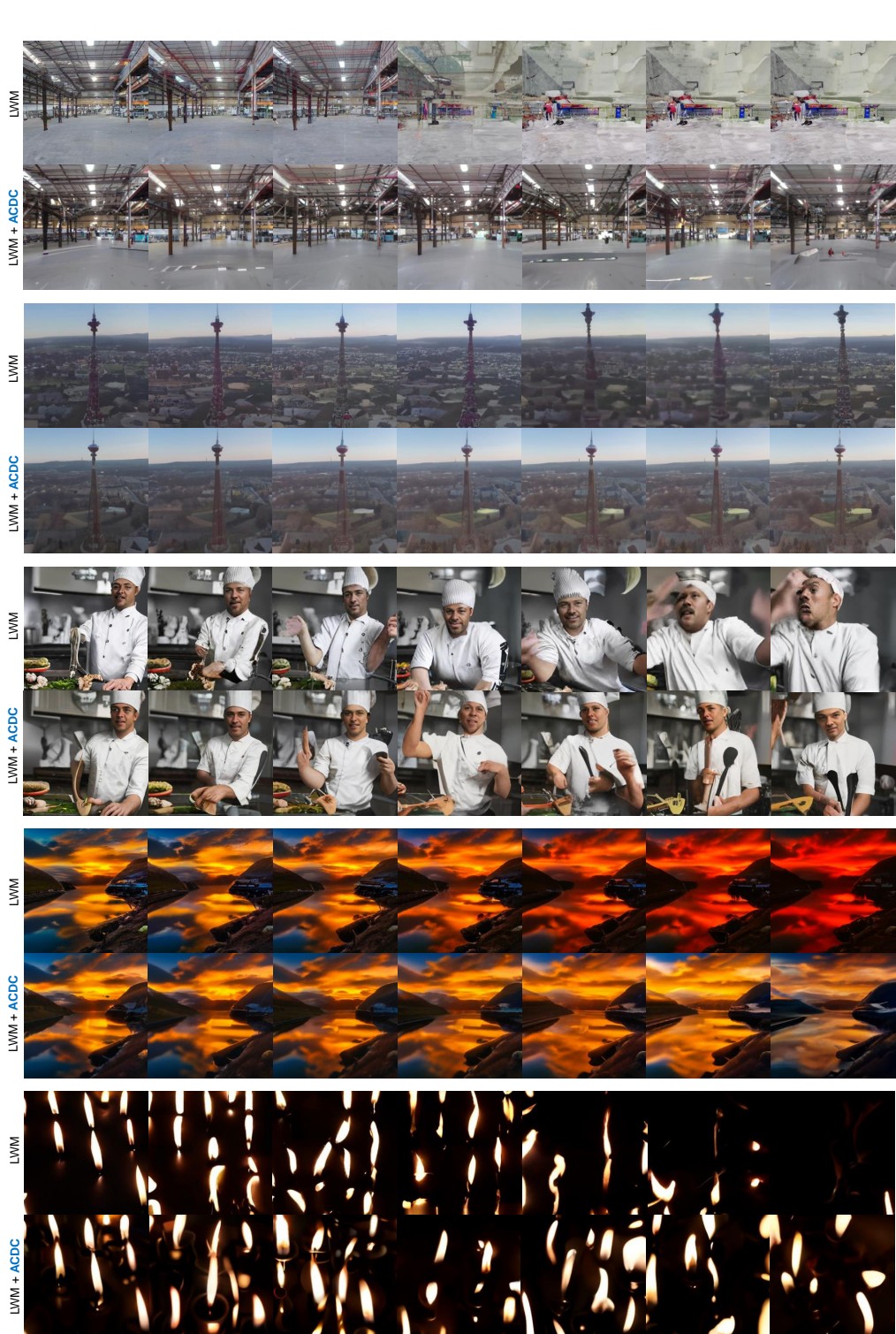

Figure 10: Further results of ACDC applied to LWM.

