# OpenReview forum: "ACDC: Autoregressive Coherent Multimodal Generation using Diffusion Correction"
_ICLR.cc/2025/Conference — ICLR 2025 Conference Withdrawn Submission_

### Official Review · Reviewer_ucrk · 2024-10-19

**Soundness:** 3
**Presentation:** 3
**Contribution:** 3
**Rating:** 5
**Confidence:** 4

**Summary:**

This paper proposes Autoregressive coherent multimodal generation with Diffusion Correction (ACDC) to levarage the  intriguing strengths of both pre-trained autoregressive models and diffusion models in a training-free manner for coherent multi-frame story generation and autoregressive video generation. The authors utilize diffusion models for local correction through SDEdit and incorporate a memory module with LLMs to address  the exponential error accumulation inherent in autoregressive models.

**Strengths:**

1. The motivation of this paper is clearly introduced through analyzing the strengths and weaknesses of autoregressive models (ARMs) and diffusion models (DMs).
2. The correction for the exponential accumulation in errors of autoregressive models for visual generation is a significant issue. The authors propose a training-free and model-agnostic approach to address this issue, and achieve improved results of coherent multi-frame story generation and autoregressive video generation.
3. The authors propose a multimodal story generation benchmark for the evaluation of multi-frame story generation.

**Weaknesses:**

1. The core technical contribution of this method is a memory-conditioned diffusion model through SDEdit, which, in my opinion, seems to lack sufficient novelty. Of course, I would like to see the authors' response and the opinions of other reviewers to assess whether my evaluation is reasonable.
2. Since the authors utilized an additional LLM memory module and diffusion correction, they should provide information on the extra inference overhead compared to the baseline model.
3. For the quantitative evaluation of the Story generation task, only the beseline model without ACDC and Stable Diffusion are evaluated. Some models specifically designed for story generation should be taken into account for evaluation, such as MM-Interleaved[1].

[1] MM-Interleaved: Interleaved Image-Text Generative Modeling via Multi-modal Feature Synchronizer

**Questions:**

Please see weaknesses.

---

### Official Review · Reviewer_qCyd · 2024-10-23

**Soundness:** 2
**Presentation:** 3
**Contribution:** 1
**Rating:** 3
**Confidence:** 5

**Summary:**

The authors try to combine ARMs and DMs into a multi-modal model. They find ARMs suffer from error accumulation, and DMs are limited by their local focus. Their ACDC addresses these issues by combining ARMs for global context and memory-conditioned DMs for local corrections, ensuring high-quality outputs. Experiments demonstrate ACDC's ability to reduce errors and improve performance across multimodal tasks, without additional fine-tuning.

**Strengths:**

- This paper is well-structured and easy to follow, with clear explanations of the proposed concepts. Figure 2 effectively illustrates the main architecture.
- One of the key advantages of the proposed approach is that it requires no additional training, which makes it highly efficient and adaptable.
- The method can be applied to a wide range of tasks, including story generation and video generation.

**Weaknesses:**

- This paper introduces diffusion correction to address visual artifacts in multimodal ARMs generation. However, state-of-the-art models, such as Emu2 (CVPR 2024), have already largely resolved this issue, which largely weakening the contribution of this work.
- The SDEdit-like method adds extra computational overhead, which may not be justified given the current advancements.
- The paper lacks sufficient experiments and comparisons. More baselines are needed to validate the effectiveness of the proposed ACDC method. For video generation, only LWM is compared, which is insufficient. For story generation, the paper should include comparisons with more works, such as ARMs-based models (StoryGPT-V, SEED-Story) and DMs-based models (ConsiStory, StoryGen).
- The improvements in video generation results are minimal, as evident from the provided page.
- In my view, the introduction lacks depth and insights. There should be a more thorough analysis of the challenges faced by both DMs and ARMs. For instance, as the paper claims, "ARMs often suffer from exponential error accumulation over long sequences, leading to physically implausible results, while DMs are limited by their local context generation capabilities."

**Questions:**

My major concern lies in the depth of the main idea and the experiments.

---

### Official Review · Reviewer_HCRo · 2024-11-04

**Soundness:** 2
**Presentation:** 3
**Contribution:** 1
**Rating:** 3
**Confidence:** 4

**Summary:**

This paper introduces a tuning-free method that combines multi-modal autoregressive models (ARMs) and diffusion models. The proposed method, termed ACDC, uses diffusion models to correct the visual outputs of multi-modal ARMs, conditioned on the LLM refined text-prompts. As demonstrated by the experiments, ACDC achieves more coherent generation results in terms of both multi-frame story generation and autoregressive video generation.

**Strengths:**

- This paper for the first time proposes a flexible zero-shot combination of multi-modal autoregressive models (ARMs) and diffusion models (DMs). With the help of DMs' correction conditioned on LLM-refined text prompts, the model effectively improves the generation quality of multi-modal ARMs.
- This paper is clearly written and easy to understand.

**Weaknesses:**

- The proposed method, i.e., ACDC, is technically trivial.
  - This method can be considered as a simple application of using diffusion techniques (e.g., SDEdit, DDIM) to correct the frame output of multi-modal autoregressive models (ARMs). I don't see any challenges (and how does this paper address these challenges) to do so.
  - For the so-called "Incorporating user constraints", it is very travail. Basically,  anyone can apply SD-inpainting to correct the generated frames of any ARMs (or world models).

- Technically, ACDC can not assure the temporal consistency of adjacent video clips, since they are corrected (re-generated) from perturbed frames and conditioned on different LLM-refined text prompts. It may even break the original temporal consistency of adjacent video clips generated by the multi-modal ARMs.
  - Since the paper only provides several frames of different text prompts, e.g., figure 4, it's unclear whether these scenes transition smoothly.
  - From my perspective, since the diffusion model's corrections are based on different perturbations, the consistency of consecutive video clips might be worse than some existing multi-text video generation methods, e.g., Gen-L-Video [1], which is also tuning-free.


- This method seems to have little correlation with multi-modal ARMs. I.e., if one wants to design methods that use LLMs and diffusion models to correct/refine the per-chunk video generation result, why it must be applied on multi-modal ARMs? There are many multi-text video generation models, e.g., [2],[3], which are capable of generating consecutive video clips described by sequential text prompts. This paper fails to compare the proposed ACDC to these methods (or apply ACDC on top of these methods), and apparently the generation results (as shown in its [project page](https://acdc2025.github.io/)) of ACDC are inferior to these methods in terms of both visual quality and temporal consistency.

[1] Wang, Fu-Yun, et al. "Gen-l-video: Multi-text to long video generation via temporal co-denoising." *arXiv preprint arXiv:2305.18264* (2023). ( https://github.com/G-U-N/Gen-L-Video )

[2] Villegas, Ruben, et al. "Phenaki: Variable length video generation from open domain textual descriptions." *International Conference on Learning Representations*. 2022.

[3] Oh, Gyeongrok, et al. "MEVG: Multi-event Video Generation with Text-to-Video Models." *European Conference on Computer Vision*. Springer, Cham, 2024.

**Questions:**

- Are there any qualitative examples to show the transition consistency of adjacent video clips (i.e., a video version of Figure 4) ?
- In Eq. (2), It seems that the $x_t^i$ in the second term should be $x_{t'}^i$.

---

### Official Review · Reviewer_23Uz · 2024-11-06

**Soundness:** 3
**Presentation:** 3
**Contribution:** 2
**Rating:** 6
**Confidence:** 3

**Summary:**

This paper proposes to leverage pre-trained text-to-image diffusion models to guide the inference process of autoregressive models for story and video generation tasks. Additional large language model is also used to refine the text prompt with clearer details. The proposed framework is able to fit in any base model to enhances them, and reaches leading performance on multi-frame story and video generation benchmarks.

**Strengths:**

- The visualizations present well surpassing generative quality of the proposed framework.

**Weaknesses:**

- How is LLM applied to (short) video generation as few frames won't have too much content changed?

- How about the size, memory and computation cost for the proposed model when it's being additionally added to the baseline? Comparisons with similar total parameters or inference time should be also conducted as this would be crucial for end users.

**Questions:**

- The FID column in Table 1 should have downside red triangle.

- It's better to show the generation result corresponding to the text prompt example in Figure. 3 etc.

---

### Note · Authors · 2024-11-13

I have read and agree with the venue's withdrawal policy on behalf of myself and my co-authors.